

# Assessment of light-absorbing carbonaceous aerosol origins and properties at the ATOLL site in Northern France

Alejandra Velazquez-Garcia[1,2], Joel F. de Brito[1], Suzanne Crumeyrolle[2], Isabelle Chiapello[2], Véronique Riffault[1]

[1]Centre for Energy and Environment, IMT Nord Europe, Institut Mines-Télécom, Université de Lille, Lille, 59000, France
[2]LOA-Laboratoire d'Optique Atmosphérique, CNRS, UMR 8518, Univ. Lille, 59000, Lille, France

*Correspondence to*: Alejandra Velazquez-Garcia (ale-vg13@hotmail.com) & Joel F. de Brito (joel.brito@imt-nord-europe.fr)





**Abstract.**

Understanding the lifecycle of light-absorbing carbonaceous aerosols, from emission to deposition, is critical for assessing their climate impact. This study integrated multi-year aerosol observations from the ATOLL (Lille metropolis, northern France) platform, with air mass back-trajectories and emission inventory as a newly developed 'INTERPLAY' approach. Applied to Black Carbon (BC), the method apportioned source contributions (shipping, vehicular, residential heating, industrial) and studied aerosol aging effects, notably on the Brown Carbon (BrC) component. Results estimate that throughout the year, vehicular traffic dominated BC (31%), followed by shipping (25%, of which one-third was from canals/rivers) and residential heating (21%). Comparing INTERPLAY results with the aethalometer model highlights that the 'residential sector' BC can be entirely apportioned to BC from wood burning ($BC_{wb}$), notably in winter, while vehicular traffic corresponds to only about 41% of BC fossil fuel ($BC_{ff}$) at the ATOLL site, the rest being apportioned to shipping (33%) and industrial (23%) emissions. Thus, vehicular traffic and $BC_{ff}$ should not be used interchangeably, particularly in regions near intense maritime traffic. Concerning BrC, our analysis confirms a dominant role of residential heating. Focusing on winter, results suggest a considerable decrease in the BrC component only 24 hours after emission, with fresh residential emissions being responsible for 72% of BrC absorption at ATOLL. Improving our understanding of sources and dynamics of light-absorbing carbonaceous aerosols is crucial for both source abatement strategies as well as a better assessment of their climate impact.

Keywords: air quality, climate impact, Black Carbon, Brown Carbon, shipping pollution



## 1 Introduction

Light Absorbing Carbonaceous (LAC) particles are an important component of the climate system due to their effect on solar radiation (Aerosol-Radiation Interactions, ari) and the associated impact on cloud properties (Aerosol-Cloud Interactions, aci) (Forster et al., 2021). In addition, deposited LACs efficiently reduce snow/ice albedo, increasing heating and accelerating melting (Pu et al. 2017; Wang et al. 2017). LACs are mainly composed of organic and inorganic components, namely Brown and Black Carbon (BrC and BC, respectively). Whereas BC absorbs over a broad spectral range from UV to infrared and is

widely recognized as an important climatic driver, BrC absorbs mostly in the UV and its impacts suffer from large uncertainties due to the complex variety of molecules or molecular aggregates dictating its optical properties, and how atmospheric processing forms or destroys those compounds (Laskin, and Nizkorodov 2015; Moise, Flores, and Rudich 2015).

To date, numerous LAC sources have been identified. While BC primarily originates from combustion processes, BrC can form in the atmosphere via secondary reactions. Once present in the atmosphere, the characteristics of LAC can evolve due to

changes in their chemical composition, as well as their physical properties, including size distribution, phase state, viscosity, etc. (Laskin, and Nizkorodov 2015; Moise, Flores, and Rudich 2015). BC typically exhibits an atmospheric lifetime of 7 to 10 days, aging rapidly within a few hours over polluted regions such as urban and biomass-burning impacted areas (Denjean et al., 2020; Kumar et al., 2018; Saturno et al., 2018; Sun et al., 2021). In contrast, its aging timescale can extend to days or even weeks in remote areas. Conversely, BrC, as a component of organic aerosol (OA), is highly prone to atmospheric processing

such as oligomerization, functionalization, or fragmentation. Sources of BrC encompass both anthropogenic, such as the combustion of fossil fuels and biomass, and biogenic sources arising from the oxidation products of vegetation-emitted terpenes/isoprene, as well as primary biological aerosol particles from plants and fungi, among others (He et al., 2021; Kanakidou et al., 2005; Ng et al., 2017). Biomass burning (BB), including forest fires and the burning of crop residues, is considered to be the main source of primary BrC (Teich et al., 2017; Lin et al., 2017). Corbin et al. (2019) found that BrC, as

highly absorbing tar balls, may dominate the total aerosol light absorption of ship emissions in the open ocean and the Arctic, due to heavy-fuel oil. At the global scale, shipping emissions represent only a minor contribution to LAC mass (Bond et al. 2013). However, they tend to occur in remote regions otherwise quite pristine and climatically sensitive (Ødemark et al., 2012). Production of secondary BrC has been shown to occur within or between the gas phase, particle phase, and cloud droplets (Laskin and Nizkorodov, 2015). Various mechanisms contribute to its formation, including the nitrification of aromatic

compounds (Lu et al., 2011; Harrison et al., 2005), the acid-catalyzed condensation of hydroxyl aldehyde followed by oligomerization (Haan et al., 2009; Shapiro et al., 2009), and the reaction of ammonia ($NH_3$) or amino acids with carbonyls (De Haan et al., 2011; Nguyen et al., 2013; Flores et al., 2014). Those processes can either enhance or decrease the light-absorbing capabilities of the OA, generally termed bleaching (Laskin, and Nizkorodov 2015). Specifically, BrC can be photolyzed and degraded into a less absorbing compound when directly exposed to solar radiation (photo-bleaching).

Accurately representing this process can prove challenging, due to variations in the photolysis rates of chromophores, the





absorbing components of the molecules (Chen et al. 2021; Hems et al. 2021; Liu et al. 2021; Liu et al. 2016). Only a few modeling studies tried to include the BC and BrC lifetimes to estimate their climate impacts (Saleh 2020).

Therefore, a better assessment of real-world LAC properties according to their sources and age is fundamentally relevant to improving our understanding of the overall impact of aerosol particles on climate. In this work, we employ an innovative
analysis approach combining multiannual submicron aerosol (PM$_1$) observations, air mass back trajectories (BTs), and emission inventories to examine how different sources and geographical origins affect the optical properties of carbonaceous aerosols at an observation site. The study is centered at the ATOLL (ATmospheric Observations in liLLe) suburban site in Northern France. The region is a densely populated pollution hotspot in Europe, with a significant fraction of secondary aerosols (Chebaicheb et al. 2023;  Chen et al. 2022;  Rodelas, et al. 2019; Zhang et al. 2021), and is strongly impacted by
LACs, having a major role in aerosol light extinction at the ATOLL site (Savadkoohi et al., 2023; Velazquez-Garcia et al., 2023).

## 2 Materials and Methods

### 2.1 Sampling site

Real-time in situ aerosol measurements have been performed routinely since October 2016 on the ATOLL platform, located
on the rooftop of a University of Lille building (50.6111 °N, 3.1404 °E, 70 m a.s.l.), 6 km southeast of Lille downtown area. The ATOLL site is part of the French ACTRIS (Aerosols, Clouds, and Traces Gases Research InfraStructure) National Facilities for both in situ and remote sensing measurements, as well as the CARA program (Favez et al. 2021) concerning the chemical composition of aerosols, both aiming at continuous, long-term, and high-quality atmospheric data. BC concentrations and absorption coefficients ($\sigma_{abs}$) are continually measured with a 1-min time resolution using a 7-wavelength (370, 420, 525,
590, 660, 880, 950 nm) aethalometer (AE33, Magee Scientific Inc., Drinovec et al., 2015; Cuesta-Mosquera et al., 2021). Following ACTRIS current guidelines (https://actris-ecac.eu/particle-light-absorption.html), $\sigma_{abs}$ coefficients at each wavelength have been recalculated by multiplying BC by the mass-specific absorption coefficient (MAC) then dividing by the suitable harmonization factor to account for the filter multiple scattering effects: 2.21 (M8020 filter tape) in 2017 and 1.76 (M8060 filter tape) for 2018 and 2019. The aethalometer is sampling at 5 L min$^{-1}$ downstream a PM$_1$ cyclone (BGI SCC1.197,
Mesa Labs) with a stainless-steel line (3.5 m) and flexible tubing (0.50 m) designed to limit the aerosol electrostatic losses. It is noteworthy that whereas absorption measurements in the PM$_1$ fraction are less prone to be affected by artifacts at higher sizes as typically deployed (PM$_{2.5}$ or PM$_{10}$), they can suffer from BC underestimation, potentially missing aged, internally mixed BC-containing aerosols. Depending on focus and statistics, subsequent analyses have been split between warm (May-Aug) and cold seasons (Sep-Apr), or restricted to summer (Jun-Aug) and winter (Dec-Feb).



## 2.2 Aerosol optical properties

The Absorption Ångström Exponent (AAE) parameter is derived from $\sigma_{abs}$ and describes the wavelength ($\lambda$) dependence of aerosol light absorption, also employed in aerosol characterization and apportionment studies (Liu et al. 2018). Here we approximate the AAE (for each datapoint) between two wavelengths ($\lambda_1 = 370$ nm and $\lambda_2 = 880$ nm) by:

$$AAE\left(\frac{\sigma_{abs,\lambda1}}{\sigma_{abs,\lambda2}}\right) = -\frac{log\left(\frac{\sigma_{abs,\lambda1}}{\sigma_{abs,\lambda2}}\right)}{log\left(\frac{\lambda1}{\lambda2}\right)} = -\frac{log(\sigma_{abs,\lambda1})-log(\sigma_{abs,\lambda2})}{log(\lambda1)-log(\lambda2)}$$

Eq. 1

The spectral dependence of BC absorption ($\sigma_{BC\ abs}$) can be derived based on the measured total aerosol absorption coefficients ($\sigma_{abs}$) using Eq. 1 assuming that BC is the only species responsible for light absorption at 880 nm. The AAE for BC usually ranges between 0.9 and 1.2 (Bond et al. 2013; Lu et al. 2015), thus we applied an AAE of 1 in our analysis. In addition, BrC absorption coefficients ($\sigma_{BrC\ abs}$) were calculated at each wavelength using Eq 2 and BrC wavelength dependence was then estimated.

$$\sigma_{BrC\ abs,\lambda} = \sigma_{abs,\lambda} - \sigma_{BC\ abs,\lambda}$$

Eq. 2

It is noteworthy that the assumptions on the AAE of pure BC ($AAE_{BC}$) to retrieve BrC absorption can impact its estimates. For instance, Zhang et al., (2020) applied a sensitivity test using three different values (0.9, 1, and 1.1) for $AAE_{BC}$, and reporting an uncertainty of 11%. Other strategies, as used by de Sá et al. (2019), for example, are based on the assumption that $AAE_{BC}$ varies over time, and can be estimated from $\sigma_{abs}$ slopes in the vicinity of 880 nm. Furthermore, we used the "Aethalometer model" approach (Sandradewi et al., 2008) for the estimation of the BC coming from fossil fuel ($BC_{ff}$) and from wood burning ($BC_{wb}$) sources. The equation relating the wavelengths (i.e. 370 and 880 nm) and the absorption coefficient for the fossil fuel and wood burning are:

$$\frac{\sigma_{abs,ff}(\lambda1)}{\sigma_{abs,ff}(\lambda2)} = \frac{\lambda1}{\lambda2}^{-AAE_{ff}}$$

Eq. 3

$$\frac{\sigma_{abs,wb}(\lambda1)}{\sigma_{abs,wb}(\lambda2)} = \frac{\lambda1}{\lambda2}^{-AAE_{wb}}$$

Eq. 4

As indicated in Equations 3 and 4, the estimates are also dependent on assumptions of $AAE_{ff}$ and $AAE_{wb}$, with typically used values of 1 and 2, respectively. Nonetheless, different values have also been proposed in the literature, based for example on Elemental Carbon [14]C analysis (e.g. Savadkoohi et al. (2023) and references therein). Due to the unavailability of [14]C analysis at the ATOLL site, and for consistency with previous studies at the site, we assumed values typically used in the literature (i.e. $AAE_{BC} = 1$, $AAE_{ff} = 1$, and $AAE_{wb} = 2$).

## 2.3 Individual back trajectory analysis

The 72-hour back-trajectories (BTs) used here were calculated for the 2016 – 2019 period with the NOAA Hybrid Single-Particle Lagrangian Integrated Trajectory (HYSPLIT) model (Stein et al., 2015). An arrival height of half the planetary boundary layer (PBL) was used at 1-hour intervals (24 trajectories per day) using the 1° × 1° resolution Global Data Assimilations System (GDAS) meteorological files.



## 2.4 Emission inventory

The emission inventory was supplied by the Emissions Database for Global Atmospheric Research (EDGAR) developed by the Joint Research Center (JRC). EDGAR provides independent estimates of global anthropogenic emissions and emission trends, based on publicly available statistics (Crippa et al. 2020). The emission inventory is characterized by a world-historical trend from 1970-2018, including emissions of all greenhouse gases, air pollutants, and aerosols. Data are presented for all countries, with emissions provided per main source category, and gridded on a spatial resolution of $0.1° \times 0.1°$. The version employed in this work (v. 6.1) is time-dependent, with temporal profiles developed for each country/region and sector-specific. Here, we chose to use monthly sector-specific grid maps for BC emissions (in kg BC/m$^2$/s, Fig. S1). Different sectors described in the inventory were grouped here for simplicity: "traffic" comprises tail-pipe and resuspension emissions; "residential" consists of heat generation either by biomass or liquid fuel; "power industry" incorporates electricity generation, heat, and power generation; "industry-manufacturing" contains industrial activities, as well as embedded electricity and heat generation, oil refineries and transformation; finally, "shipping" includes oceanic and in-land, such as rivers and canals, transportation.

## 2.5 INTERPLAY approach

The INTERPLAY (IN-siTu obsERvations, hysPLit, And emission inventorY) approach derives from the so-called footprint analysis (e.g. Pöhlker et al. 2019), which combines BTs and land information to explore air mass history and better interpret the atmospheric composition and its properties. The main advantage consists of the use of computationally inexpensive Lagrangian modeling, being able to gain important insights from large datasets, such as multi-annual observations. INTERPLAY consists of using emission inventory information to integrate the source strength of a given species along individual BTs to quantify the relative contributions to the receptor site (including information such as distance, and traveled time). Here, INTERPLAY has been applied to BC, given that it is expected to be relatively stable in the atmosphere within the time frame of BT considered here (72 hours).

The INTERPLAY method is a multi-step process that first consists of interpolating each hourly HYSPLIT trajectory (comprising 72 points corresponding to 72 hours) into a more finely resolved spatial BT. In this refined BT, each interval between two consecutive points corresponds to 10 minutes, resulting in a total of $72 \times 6$ points. For points located more than 40 km away from the receptor site, the EDGAR emissions are integrated over a square area of $0.5° \times 0.5°$, centered on the BT location. For smaller distances (< 40 km), INTERPLAY integrates only the emissions within the air mass arrival sector. This sector is defined as a quarter-circle area with a line of symmetry towards the incoming direction (Fig. S2). This criterion is established to more accurately represent emissions in the vicinity of the site, excluding contributions from locations such as Lille, located in the NW direction from ATOLL when air masses approach from the SE, and vice-versa. To reduce the bias of aerosol removal by precipitation (wash-out), only BTs with less than 1 mm of integrated rain over the 72-hour period are considered, resulting in the exclusion of about 41% of the BTs. Despite its simplistic approach, which does not consider factors such as dilution, dry deposition, and potential inaccuracies in both BT and emission inventories, the method demonstrates a





fairly good correlation with hourly averaged in-situ BC data (Pearson coefficient R = 0.41). To minimize the effect of sporadic local sources on our analysis, INTERPLAY results were aggregated into 12-hour data points, and the median of each INTERPLAY output (e.g., the relative contribution of BC sources, distances, and ages) was then calculated. These values were

compared with the 5th percentiles of 12-hour data from in-situ BC mass concentrations, i.e., the BC baseline. With this approach, the correlation between INTERPLAY and in situ BC observations was further improved, yielding a Pearson coefficient R of 0.60 (Figure S3). For context, the 1-year (2019) daily averages of BC at ATOLL exhibited a Pearson coefficient of 0.81 when compared with a computationally expensive, 3D mesoscale atmospheric model (Chebaicheb et al., in preparation). Conversely, a comparison of in-situ BC observations with FLEXPART product (https://flexpart-request.nilo.no)

using ECLIPSEv6 (Klimont et al., 2017) and GFEDv4 (Giglio et al., 2013) emission inventories yielded a Pearson coefficient R of 0.52.

The INTERPLAY method has been used to link a specific in-situ optical property with a BC source or region. This association requires a minimum contribution of 20% of the total integrated BC (e.g., traffic) for individual BTs. Statistical robustness is ensured by focusing exclusively on sources and regions represented by at least 70 BTs. The development of INTERPLAY has

been carried out using Matlab 2020a, and country apportionment was facilitated by the climate data toolbox (Greene et al., 2019). Further details for each step of the analysis are provided in Table 1.

**2.6 Estimation of BrC atmospheric lifetime**

The analysis of BrC lifetime is performed using Multiple Linear Regression (MLR), with in-situ absorption coefficients for

BrC ($\sigma_{BrC}$) being the explained variable, and the LAC contributions from the traffic and residential sectors (obtained from INTERPLAY), as dependent variables. Furthermore, to study the atmospheric dynamics of BrC, INTERPLAY enables the separation of BC components based on the time since emissions, distinguishing between "fresh" emissions (occurring less than 24 hours before reaching the site), and "aged" ones (more than 24 hours old). The focus on traffic and residential emissions is motivated by several factors: i) they have a well-established associated BrC component (typically part of the "Aethalometer

model"), with BrC linked to wood combustion for residential heating; ii) a reduced number of dependent variables has shown higher robustness on the MLR analysis; iii) it is generally accepted that traffic and residential heating are the main sources of BC in Central Europe (e.g. Savadkoohi et al., 2023). Various tests were conducted using different times since emission (6, 12, 24, and 36 hours). The separation into two categories (below and above 24 hours) proved to be robust, and aligns with literature findings, reporting a typical BrC lifetime of approximately 1 day (Forrister et al. 2015; Wang et al. 2016; Wong et al. 2019;

Saleh 2020). Detailed test results are presented in the SI (Figures S10 and S11). Calculations were performed using the least square method, with coefficients constrained to be positive, and the linear intercept set to zero to explain the absorption measured by in situ observations. The MLR analysis focused on the 12-hour averaged integrated BrC mass observed during the cold period (715 data points) and yielded a Pearson coefficient of R = 0.80 between observed and modeled BrC.





**Table 1 Statistics of observations and INTERPLAY at the ATOLL site from December 10th 2016 till December 31st 2019.**

| Method | | Parameter used | Raw Time resolution | Raw data | 12-hour baseline | Cold period (12h) | Warm period (12h) |
|---|---|---|---|---|---|---|---|
| **Aerosol in situ properties** | Aethalometer | $\sigma_{abs}$ & BC | 1 min | 1 226 414 | 1 581 | 715 | 685 |
| **Back trajectory** | HYSPLIT v. 2020 | BT | 1 hour | 11 607 864 | N/A+ | | |
| **Emission Inventory** | EDGAR v. 2018 | BC sector-specific grid maps | 1 month | 64 800 00 | | | |
| **INTERPLAY approach** | | Integrated BC | 1 hour | 6 144 703 | 675 913 | 309 595 | 293 141 |

+N/A = not applicable. The 12-hour baseline applies only for in situ observations and BC footprint from INTERPLAY for comparison.

## 3. Results and discussion

### 3.1 Wind direction and BC temporal variability

Figure 1 shows the wind roses and the contour plots of average concentrations of $BC_{ff}$ and $BC_{wb}$ across different wind sectors and hours of the day during the warm (May-Aug) and cold seasons (Sep-Apr), corresponding to potentially high usage of wood combustion for residential heating. During the cold period, the highest wind frequency comes from the southwest sector, with a median wind speed of 1.5 m s$^{-1}$. During the warm period, although the southwest sector remains dominant, with wind velocities comparable to the cold season, there are additional winds from the north and northeast sectors with moderate speeds

(2 – 4 m s$^{-1}$).

Across the multi-annual dataset, the average concentrations for $BC_{ff}$ and $BC_{wb}$ are 0.75 and 0.27 µg m$^{-3}$, respectively. Daily concentration analysis shows $BC_{ff}$ peaking (> 0.8 µg m$^{-3}$) during morning rush hours in both cold and warm periods across all wind sectors, highlighting the impact of local emissions, overlaid on a background $BC_{ff}$ of about 0.2-0.5 µg m$^{-3}$ throughout the day. In contrast, the evening peak is less pronounced during the warm period, likely due to a higher boundary layer height,

enhanced dispersion associated with warmer temperatures, lower emissions from residential heating in the afternoon, and a less distinct evening peak attributed to summer holidays (Jul-Aug in France). $BC_{wb}$ exhibits higher concentrations over the eastern sector, associated with continental air masses loaded with wood combustion. As expected, $BC_{wb}$ levels are significantly



higher in the cold season than in the warm season, reflecting widespread wood combustion for heating (Favez et al. 2021). Notably, despite the proximity of ATOLL to Lille (~4 km), the wind direction from the city (NW) does not show particularly

high $BC_{ff}$ or $BC_{wb}$ levels, indicating a more regional contribution from the densely populated area and Long Range Transport (LRT) leading to a consistently high background level. Figure S4 shows HYSPLIT backward trajectory ensembles for the studied period using air mass residence time maps, emphasizing the influence of large-scale trade wind circulation at the ATOLL platform. Similarly, consistent with a recent four-year study at the ATOLL site (Chebaicheb et al., 2023), the highest concentrations of most OA species occur when ATOLL is not influenced by the Lille city center. This study identified that

OA was mainly composed of aged, oxidized compounds (74%), with only a small fraction related to fresh traffic and wood combustion (11% and 14%, respectively). Furthermore, during winter, residential heating (combining fresh and aged components) accounts for about 50% of the OA. This corroborates that ATOLL is not dominated by local emissions, but rather by regional and LRT. Another analysis combining offline organic tracer analysis and online instrumentation (Zhang et al. 2020) demonstrated a strong correlation between BrC absorption at 370 nm and levoglucosan (a molecular tracer of cellulose

pyrolysis) across nine air quality monitoring networks in France during winter, suggesting a substantial contribution of wood burning emissions to ambient BrC aerosol. Specifically, at a suburban site in Paris, the main BrC contributors were found to be both biomass-burning OA factors. In a recent analysis, Velazquez-Garcia et al. (2023) underlined the significant role of LAC at the ATOLL site, estimating that organics are responsible for 22% of absorption in the UV range, while BC and OA contribute to 50% of the extinction in the visible range (525 nm). The strong contribution of LAC and BrC at ATOLL makes

the site suitable for an in-depth study of their sources and dynamics, as presented in the following sections.



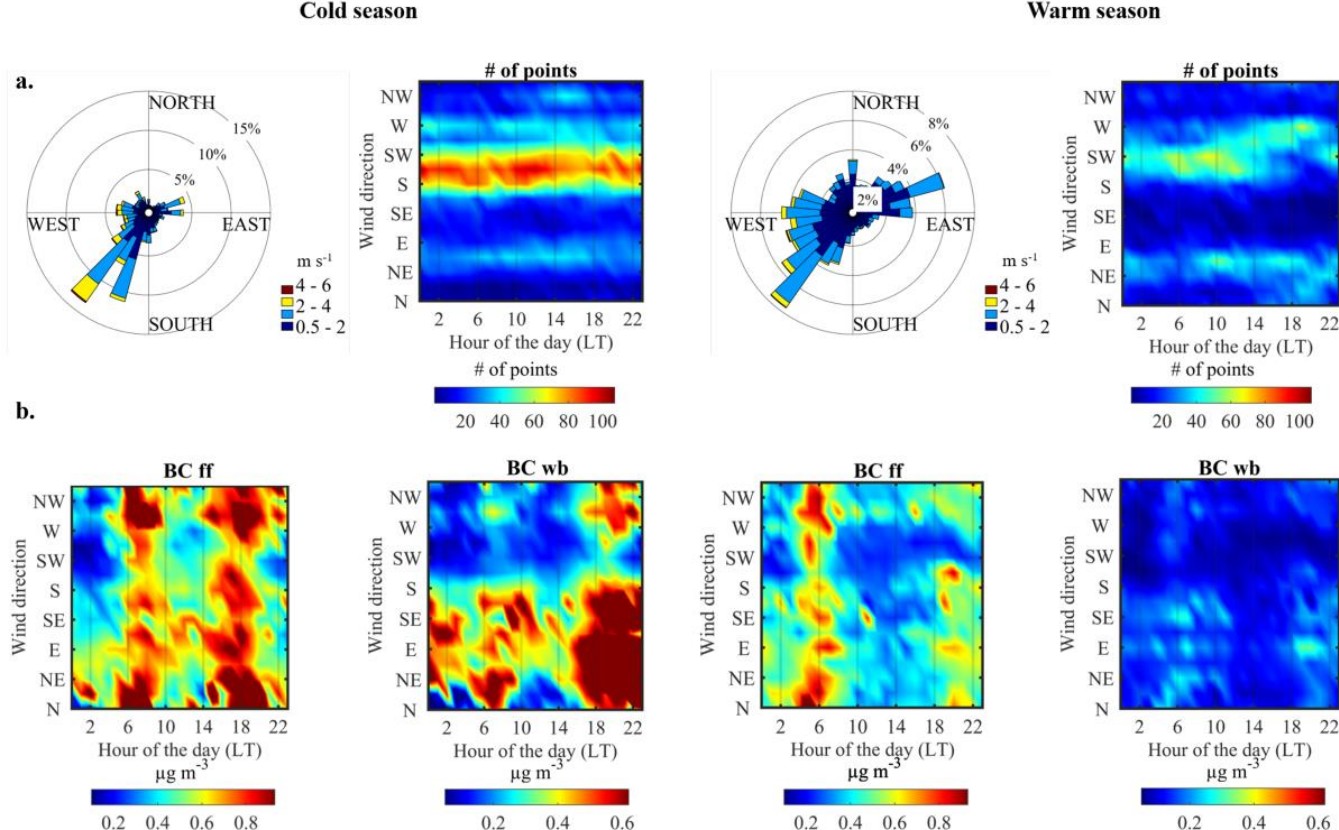

**Figure 1: Wind analysis and BC$_{ff}$ and BC$_{wb}$ levels at the ATOLL site during the cold (left, Sep-Apr) and warm (right, May-Aug) seasons. a) Wind roses and wind frequency by the hour of the day (minimum wind speed of 0.5 m s$^{-1}$). b) Contour plots of BC component loadings according to the wind sector and hour of the day. The color scale minimum and maximum values in (b) correspond to the 5$^{th}$ and 75$^{th}$ percentiles for each season, respectively.**

## 3.2 Contributions of sectorial sources and geographical origins to BC at ATOLL

Figures 2a and 2b depict the spatial apportionment of BC reaching the ATOLL platform, presenting source strengths multiplied by back trajectory frequency at a given grid point (2a) and contributions by country/region (2b). Both figures show that, aside from local sources (Lille area: 8%), most of the BC impacting the site originates from France (26%, including the Paris area) and the Benelux countries (Belgium 13%, the Netherlands, Luxemburg), followed by Germany (8%), the United Kingdom (7%, with the London area particularly highlighted), and shipping lanes in the English Channel (6%) and the North Sea (9%). Altogether, these origins account for 77% of the total BC (with "others" representing additional European countries and maritime areas). These results align with previous findings on PM$_{10}$ source regions in northern France, utilizing trajectory-based statistical models (TMs) (Waked et al., 2018) or a combination of TMs, chemical transport models and in situ observations (Potier et al., 2019).

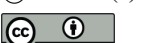



In Figure 2c, the contributions from different emission sectors from INTERPLAY are presented, as well as in-situ derived $BC_{ff}$

and $BC_{wb}$ based on the aethalometer model. INTERPLAY results indicate that the majority of BC at ATOLL throughout the

year originates from traffic (31%), followed by shipping (25%, including both sea/oceanic - 15% - and inland waterways -

10%), residential (21%), and industry-manufacturing (17%). Agricultural waste burning (4%) and the power industry (2%) are

minor contributors to BC at ATOL L. Figure S5 illustrates the geographical distribution of the sectors. Interestingly, a

comparison with in-situ observations shows a remarkably comparable contribution of biomass-burning related BC, specifically

from residential emissions and agricultural waste burning (25%), with $BC_{wb}$ (26%). The attribution of shipping at ATOLL,

exhibiting a spectral signature closer to $BC_{ff}$ rather than $BC_{wb}$ (as discussed in Corbin et al.; 2019, for example) is supported

by several aerosol observations in Dunkirk harbor using aerosol mass spectrometry and source apportionment (Zhang et al., in

preparation; Murari et al., in preparation), as well as by INTERPLAY analysis in section 3.3. Conversely, the results suggest

that $BC_{ff}$ is not directly attributable to traffic emissions. According to our analysis, traffic corresponds only to 41% of $BC_{ff}$ at

the ATOLL site, with the remainder attributed to shipping (33%), industry-manufacturing (23%), and a minor contribution of

<3% from power generation. This finding warrants care when using $BC_{ff}$ (also termed BC liquid fuel, $BC_{lf}$) from the

aethalometer model interchangeably with traffic, particularly in regions with intense shipping activities, whether maritime or

in canals.

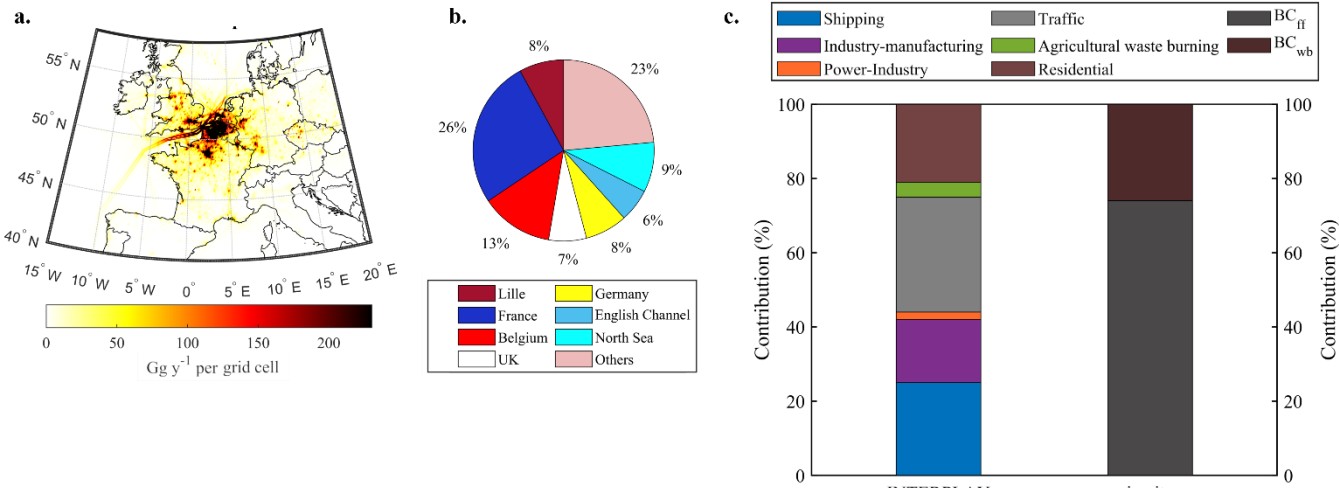

**Figure 2: (a) Spatial contribution of BC to ATOLL in Gg y$^{-1}$ per grid cell and its (b) relative contribution by region/country; c)**
**sectorial distributions according to INTERPLAY and in-situ via the aethalometer model. Results over the period Dec 2016 – Dec**
**2019.**

Figure 3 shows the spatial distribution of BC and its relative distribution of sources and origins during summer (JJA) and

winter (DJF). Spring and fall spatial distributions exhibit mixed patterns between the other contrasting seasons and are

therefore only shown in the supplementary material (Figure S5). Results show a relatively constant fraction associated with

local emissions (Lille) across all seasons (7-9%), as well as contributions from France and Belgium, corresponding generally



to about half of BC. The main geographical difference between seasons lies in the shipping contribution from the English Channel and the North Sea. Focusing solely on this sector, the North Sea accounts for 34%, followed by the English Channel with 22% (Figure S6). Surprisingly, shipping is the predominant source of BC at ATOLL during summertime (38%), with

12% apportioned to inland waterways and 24% to maritime emissions in the Channel and North Sea, despite the site being located 70 km from the coast, and 90 km from Dunkirk harbor. Following shipping, the major contributors during summer are traffic (33%), and industry-manufacturing (17%) with a minor contribution from residential sources (8%), as expected for the season. The significant summertime contribution from shipping to BC is driven by increased activity (Jalkanen et al. 2016), and more frequent favorable winds (Figure 1) from the maritime sectors (North Sea and English Channel) combined with a

decrease of vehicular traffic during summer holidays (Chebaicheb et al., 2023). Indeed, the southern North Sea, housing major harbors in Rotterdam, Antwerp, Hamburg, Zeebrugge, and Dunkirk, experiences one of the highest ship traffic densities in Europe (Eurostat 2022). Corbett et al. (1999) reported that nearly 70% of ship emissions occur in areas with potentially adverse effects on air quality inland. Liu et al. (2017), combining field measurements and modeling, estimated that ship emissions contribute 20-30% (2-7 µg m$^{-3}$) to $PM_{2.5}$ within tens of kilometers of coastal and riverside Shanghai during ship-plume-

influenced periods.

During summer, in-situ data yields a $BC_{wb}$ contribution of 16%, somewhat higher than residential and agricultural waste burning (9%), possibly explained by unaccounted forest fires, not considered in INTERPLAY. Conversely, reflecting the impact on organic aerosol (Chebaicheb et al., 2023), residential emissions dominate BC sources during winter (38%), followed by traffic (28%), industry-manufacturing (16%), and shipping (13%). The power industry and agricultural waste burning

contribute less than 5% across all seasons. $BC_{wb}$ during winter closely mirrors the combined contribution of residential and agricultural waste burning from INTERPLAY (37% and 40%, respectively). Across both seasons, traffic accounts for only about 40-46% of $BC_{ff}$, indicating that the latter should not be solely apportioned to the former, notably in regions with intense shipping activities.



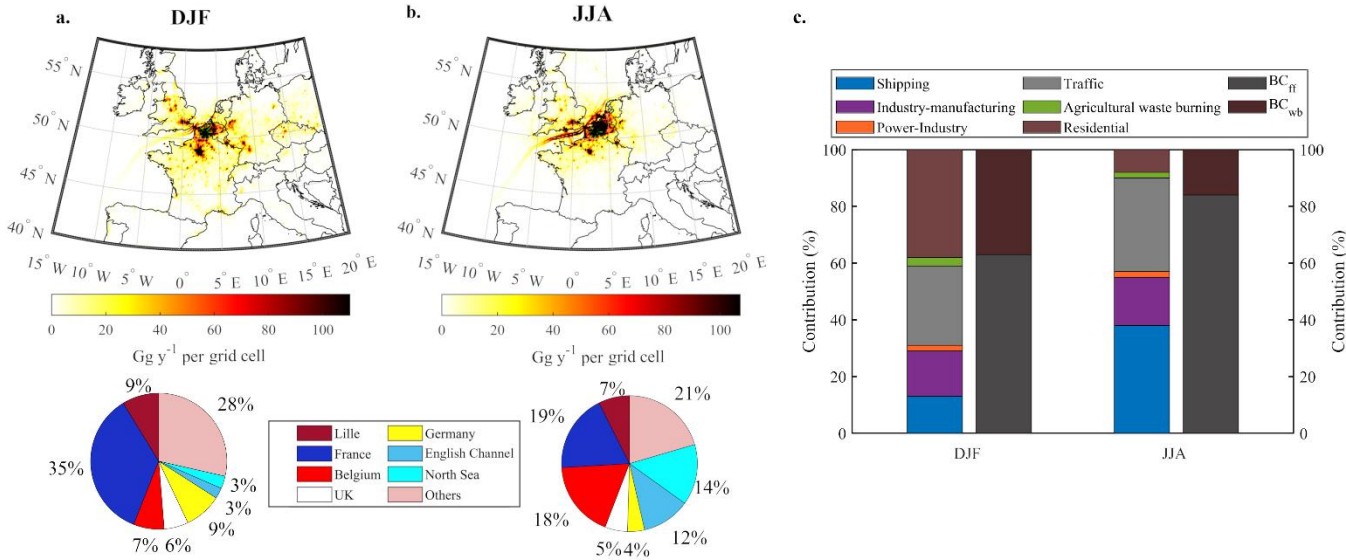


**Figure 3: Spatial contribution of BC to ATOLL in Gg y$^{-1}$ per grid cell and its relative contribution by region/country for winter (a) and summer (b); c) sectorial distributions according to INTERPLAY and in-situ via the aethalometer model. Results over the period Dec 2016 – Dec 2019.**

## 3.3 Sectorial absorption spectral signature

In addition to assessing the geographical and source contributions of LAC observed at ATOLL, one of the interesting features of INTERPLAY is its ability to link additional in situ aerosol properties from observations with various BC sectorial sources (e.g., traffic, residential, shipping, etc.). As outlined in the methodology section, by using the in-situ observations and INTERPLAY to filter data when the incident air mass comprises at least 20% BC from a given source, we computed the averaged $AAE_{370-880nm}$ as well as concentrations of $BC_{ff}$ and $BC_{wb}$ from the spectral dependency. Results of the $AAE_{370-880nm}$ filtered by source type and region are shown in Figure 4a and Figure 4b, respectively. In Figure 4a, the residential sector exhibits an average $AAE_{370-880nm}$ of approximately 1.5 (with values up to 1.6), whereas industry-manufacturing, traffic, and shipping emissions have a lower average $AAE_{370-880nm}$ of about 1.3. The comparable $AAE_{370-880nm}$ values among these three sources reinforce their categorization into $BC_{ff}$ apportionment, as discussed in the previous section. Furthermore, the literature reports a range of AAE values for industry and shipping emissions. Helin et al. (2021) note that for the industry sector, $AEE_{370-950nm}$ ranges from <0.7 when using pure coal to up to 1.3 when considering a mixture of coal and wood pellets. Conversely, for shipping, $AAE_{370-950nm}$ was observed up to 2.0 when high-sulfur content heavy fuel oil was used, compared to 1.4 when burning low-sulfur fuel. The data from ATOLL agrees with a lower AAE range for shipping, suggesting a prevalence of low sulfur content fuels in the region, consistent with the fact that the North Sea and the English Channel are both sulfur emission-controlled areas (Jonson et al., 2020).



Interestingly, the obtained $AAE_{370-880nm}$ values do not align with the typically used aethalometer model (1 and 2 for $AAE_{ff}$ and $AAE_{wb}$, respectively), for any of the sources (including traffic). Moreover, the values do not necessarily correspond to those identified in the literature ($BC_{ff}$ 0.9-1.1, and $BC_{wb}$ 1.6-3.5, (Savadkoohi et al., 2023). This discrepancy can be attributed to the high variability of AAE, influenced by factors such as the combustion appliance, burned material, burn cycle phase, and time

since emission. The AAE values of 1.3 and 1.6 for $BC_{ff}$- and $BC_{wb}$-associated sources could stem from source mixing within INTERPLAY itself (given the 20% threshold), or be influenced by secondary processes, leading to BrC formation and/or a different mixing state (BC with non-absorbing coatings) (Chylek et al. 2019; Moise, Flores, and Rudich 2015; Saleh et al. 2013; Zhang et al. 2020).

Figure 4c-d shows the in-situ aerosol concentration associated with the main source regions for BC (France, Belgium,

Germany, UK), separated into cold and warm seasons. The concentration of $BC_{ff}$ is higher during the cold season across all regions, likely due to a shallow boundary layer, reduced dispersion, and increased emissions from the residential sector. When air masses originate from the east (e.g., Germany), the highest concentration of both $BC_{ff}$ and $BC_{wb}$ are observed during the cold season. This is in line with previous findings, suggesting that these areas, known for being BC hotspots (e.g. transport, intensive agriculture, heavy industry, metallurgy, and exploitation of open mines), are generally associated with drier and

colder air masses, leading to lower dispersion and favoring the accumulation of air pollutants (Asmi et al., 2011; Barnaba et al., 2011; Bovchaliuk et al., 2013; Giles et al., 2012; Waked et al., 2018). This contrasts with the UK, which, despite being a significant source of BC (Fig. S1), is generally accompanied by northerly winds that promote pollutant dispersion. During the warm period, both concentration and AAE values are generally lower than in the cold seasons, due to higher dispersion and reduced residential emissions. While differences across regions are less pronounced, the overall trend remains comparable to

the cold season.

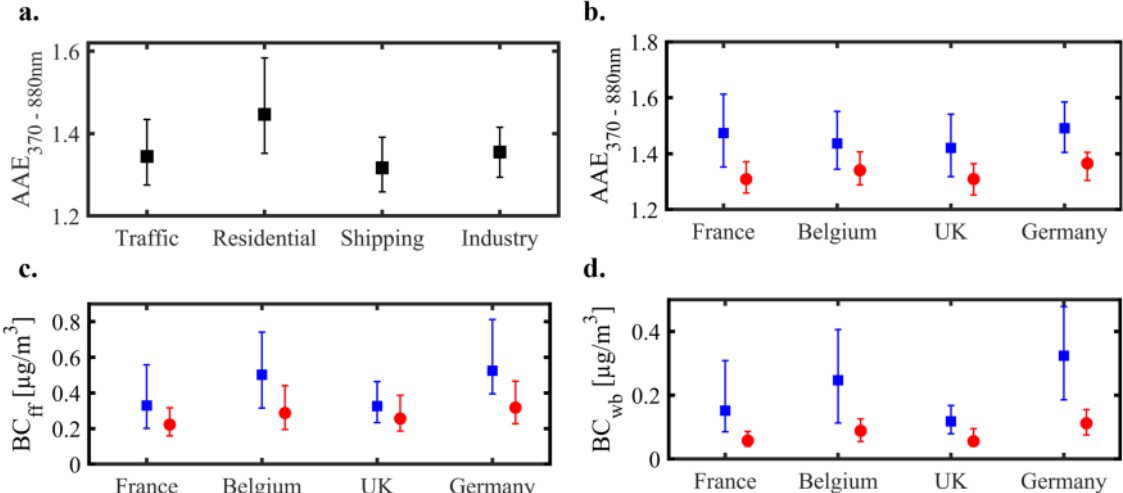

**Figure 4: Median value (markers) and interquartile range (bars) for Absorption Ångström Exponent calculated using the pair of wavelengths 370 and 880 nm ($AAE_{370-880nm}$), associated with main sectorial sources (a) and regions (b) during the cold (blue squares) and warm seasons (red circles). $BC_{ff}$ (c) and $BC_{wb}$ concentrations (d) in µg m⁻³ associated with main regions during the cold and**

**warm seasons, using the INTERPLAY method.**



### 3.4 Brown carbon origins and lifetime

Over the studied period, the average BrC absorption coefficient at ATOLL was 5.1 Mm$^{-1}$ at 370 nm, representing 22% of the total absorption at that wavelength (Velazquez-Garcia et al., 2023). This fraction is comparable to Athens (24%), despite a

higher total absorption at 370 nm (15.9 Mm$^{-1}$) (Liakakou et al. 2020) despite differences in weather conditions and heating fuel usage (Rehfeldt et al., 2020). Outside Europe, various sites report similar or higher contributions of BrC to aerosol light absorption, such as the southeastern margin of the Tibetan Plateau (20% to 40%) (Wang et al. 2019), Central Amazon (17-26%) (Saturno et al., 2018), the northern peninsular Southeast Asia (46%) (Pani et al., 2021), and in Xi'an, China (41%) (Zhu et al., 2021). Such significant differences (>40%) may be attributed to active fires over a given region, reflecting substantial

open biomass burning activities in and around the sampling location (Pani et al., 2021), along with some contribution from secondary BrC (Wang et al. 2019). At the ATOLL site, the contribution of BrC varies markedly from the warm to the cold season (from 22 to 80 %), driven by wintertime residential emissions (Chebaicheb et al. 2023; Velazquez-Garcia et al. 2023). Figure 5 shows the diel cycles of BrC absorption at 370 nm and the ratio between BC and BrC absorption during winter and summer. Overall, BrC decreased by 44% as the day progressed, likely explained by the rising boundary layer height and the

increasing rate of photobleaching/volatilization as temperatures and solar radiation peak, leading to a more efficient mixing of plumes from different sources. During winter the diel profiles are consistent with those typically observed for BC$_{wb}$, primarily linked to primary emissions (Chebaicheb et al., 2023; Chen et al., 2022; Zhang et al., 2019b, 2020b), with higher values during the late evening to early morning hours. In summer, in addition to wood combustion, BrC can be related to secondary formation through nighttime oxidation processes. Different chromophores emitted from various sources and/or formed in the atmosphere

contribute to BrC. Laboratory studies (Lin et al. 2015; Flores, and Rudich 2015; Teich et al. 2017) indicate that anthropogenic VOCs (benzene, toluene, phenols, and polycyclic aromatics hydrocarbons – PAHs) react with nitrogen oxides and produce nitro-aromatics. Several studies in urban areas (Liakakou et al. 2020; Satish et al. 2017; Stanaway et al. 2018; Wang et al. 2018; Zhang et al. 2019; Zhang et al. 2020) show a similar trend, with higher values usually observed at night. However, other authors (Gao et al. 2022; Wang et al. 2019) show an opposite trend at remote sites, where higher values are observed during

the day, attributed to the influence of photooxidation processes forming BrC through photochemical oxidation. Generally, BrC levels in the atmosphere result from a complex interplay of primary sources and secondary processes (condensation, volatilization), mixing with other aerosol types (core-shell mixing), and effects from local meteorology and boundary-layer dynamics (Lambe et al. 2013; Li et al. 2018; Romonosky et al. 2016).

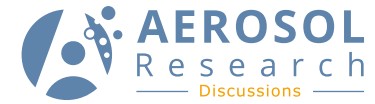

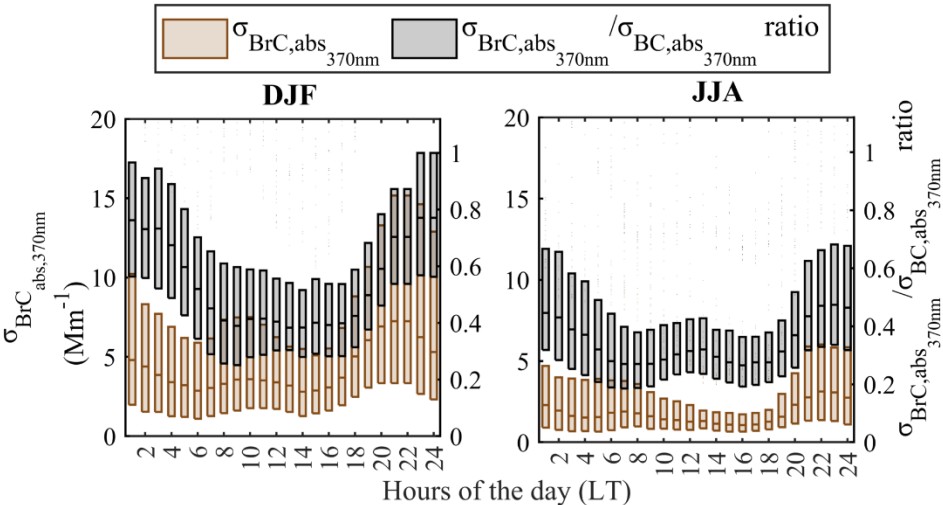


**Figure 5: Diel cycles of BrC absorption coefficient calculated at 370 nm ($\sigma_{BrC\ abs,\ 370}$ in Mm$^{-1}$) (y-axis left) and BrC fraction (y-axis right) at the ATOLL platform for the cold (left plot) and warm seasons (right plot). The line is the median, whereas the box limits correspond to the interquartile range.**

By combining in situ observations with INTERPLAY as detailed in Section 3.2 and integrating the air mass travel time since emission (i.e., age by source), we can explore the impact of atmospheric aging on BrC lifetime. This investigation is specially focused on the cold season, given the higher loadings, and the distinct identification of BrC sources due to the widespread use of wood combustion for residential heating during that period. As described in Section 2, we further differentiate the traffic and residential sources into "fresh" (BC emitted less than 24 hours ago) and "aged" (more than 24 hours ago), focusing

exclusively on the contribution of INTERPLAY 'traffic' and 'residential' sources.

Relative contributions from residential and traffic emitted less than 24 hrs (fresh) and more than 24hrs (aged) before reaching the ATOLL site: (a) the mass loading of BC; (b) the light absorption coefficient of BrC calculated at 470 nm. Figure 6a shows the relative contributions of fresh and aged mass loadings (from INTERPLAY calculations) of BC from traffic and residential sources during the cold season. It shows a relatively balanced contribution among the four categories, with fresh contributions

being slightly higher for both sectors (28%), while aged BC is higher for residential sources than for traffic (24% and 20%, respectively). Figure 6b depicts the results of Multiple Linear Regression, specifically identifying the relative contribution of these four categories of BC to in-situ BrC. The dominating component is fresh residential (72%), followed by aged residential (16%) and aged traffic (12%), with no significant contribution from fresh traffic. This analysis suggests that, after 24 hours of atmospheric aging, the BrC components associated with residential heating undergo significant depletion. The data obtained

here implies that despite the generally low photochemistry during wintertime, there is a substantial reduction in the BrC component from residential heating in this season. Table S1 provides the MAE values retrieved for the BC components through MLR. Notably, the MAE of BC from the aged residential sector decreases by a factor of 40 compared to fresh residential.





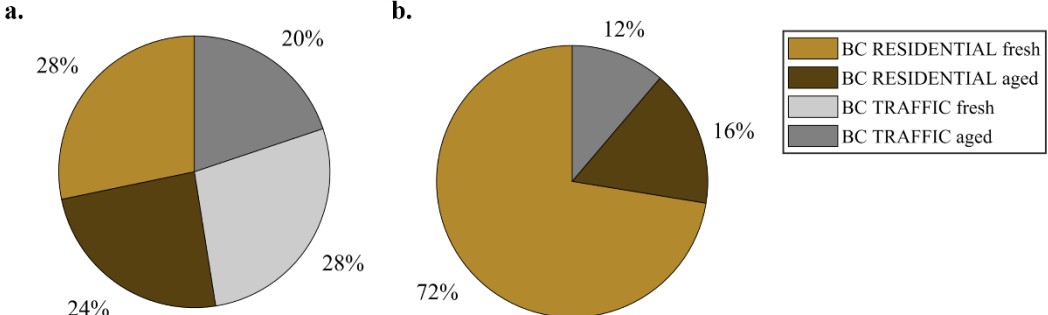

**Figure 6: Relative contributions from residential and traffic emitted less than 24 hrs (fresh) and more than 24hrs (aged) before reaching the ATOLL site: (a) the mass loading of BC; (b) the light absorption coefficient of BrC calculated at 470 nm.**

## 4 Conclusions and perspectives

In this paper, we investigated the local, national, and transboundary origins of light-absorbing carbonaceous aerosols influencing the ATOLL platform, as well as their associated spectral dependency. Our approach involved a comprehensive analysis of local winds, and we introduced an innovative method, named INTERPLAY, which combines in situ observations, back trajectories, and emission inventories. INTERPLAY is a computationally efficient tool, complementing in situ observations, particularly within a multi-year database. However, certain limitations are inherent in INTERPLAY. Notably, trajectory calculations from Hysplit, a component of the method, carry a known 15-30% error (Stein et al., 2015). Additionally, uncertainties associated with emission inventories can influence sectorial and regional contributions. For example, van der Gon et al. (2015) reported uncertainties in national emission estimates due to gaps in reported emission sources, particularly underestimations for OA over regions dominated by residential wood combustion. Discrepancies in inventories can also arise from variations in spatial grid methodologies, stemming from differences in model configurations, methodological assumptions, and weighting methodologies (van der Gon et al. 2015; Thunis et al. 2021). Finally, it is important to note that INTERPLAY does not account for dispersion, dilution, or deposition, and therefore cannot calculate atmospheric concentrations. Instead, it provides integrated mass and relative contributions. Despite these limitations, the correlation between INTERPLAY results and in-situ observations for relatively inert aerosol species, such as BC, over the studied 72-hour period, has yielded only moderately lower correlation values compared to 3-D atmospheric modeling at ATOLL for 2018 (Pearson 0.60 vs. 0.81) (Chebaicheb et al. in preparation). The precision of INTERPLAY in analyzing reactive/secondary atmospheric species is a topic for future exploration and testing.

INTERPLAY has been employed in this study to identify the sources of air masses (both regional and sectorial) reaching the ATOLL platform. This information has been used to examine their respective spectral dependency (AAE$_{370-880nm}$), compare the apportionment of BC sources (BC$_{ff}$, BC$_{wb}$), and explore the BrC lifetime and origins since the time of emission, distinguishing air masses as either fresh (emitted less than 24 hours ago) vs. aged (more than 24 hours ago) BC aerosols. The key findings can be summarized in four points:



a. Approximately 56% of BC at ATOLL originates from France, Belgium, and shipping in the Channel, while the UK and Germany contribute around 8% each. Another minor region, mostly associated with Eastern Europe, contributes about 25%.

b. The sectorial analysis reveals a surprisingly high contribution of shipping to ATOLL, with a yearly average of 25%, rising to 38% during summertime, despite the site distance from the coast (~70 km). In contrast, the residential source has the highest contribution during wintertime (38%), decreasing to 8% in summer. Traffic and industry-manufacturing contributions remain relatively constant throughout the seasons at 28%-33% and 16%-17%, respectively.

c. Comparing INTERPLAY with the aethalometer model based on in-situ data demonstrates excellent agreement in the relative contributions to residential heating and agricultural waste burning with $BC_{wb}$ (25% vs. 26%, respectively). However, results show that traffic corresponds to only about 40% of $BC_{ff}$ at ATOLL, a suburban site. This result underscores the need for caution when apportioning $BC_{ff}$ in other locations, especially in regions with intense shipping activities, without a thorough BC source apportionment analysis.

d. In situ BrC absorption has been analyzed using INTERPLAY during the cold season, confirming the expected dominance of residential emissions, primarily from fresh emissions (<24h). This suggests a significant decrease in the BrC component after just 24 hours, even during wintertime Europe.

While numerous studies have investigated BrC lifetime based on burning material and atmospheric aging, the currently existing data arise mostly from laboratory-controlled experiments or summertime events associated with wildfires. Therefore, studies focusing on the ubiquitous use of wood combustion for residential heating, particularly in Central Europe, have been lacking to our knowledge. This not only has climatic implications but also direct consequences on in situ source apportionment methods (i.e. using AAE to retrieve $BC_{ff}$ and $BC_{wb}$), potentially leading to the misattribution of 24-hour aged $BC_{wb}$ to traffic and, consequently, less efficient pollution abatement strategies.

**Code and data availability**

The script and data will be made available upon request.

**Author contribution**

Conceptualization: JFB, SC, IC, VR; Data curation: AVG, SC, VR, JFB; Formal analysis: AVG, JFB, SC; Funding acquisition: AVG, IC, VR; Investigation: AVG, JFB, SC, VR; Methodology: JFB, SC; Project administration: IC, VR; Resources: VR; Software: AVG, JFB, SC; Supervision: IC, VR; Validation: JFB, SC, VR; Visualization: AVG; Writing – Original Draft: AVG, JFB; Writing – Review & Editing: SC, IC, VR.

**Completing interests**

The authors declare that they have no conflict of interest.



**Funding sources**

IMT Nord Europe and LOA acknowledge financial support from the Labex CaPPA project, which is funded by the French National Research Agency (ANR) through the PIA (Programme d'Investissement d'Avenir) under contract ANR-11-LABX-0005-01, and the CLIMIBIO and ECRIN projects, all financed by the Regional Council "Hauts-de-France" and the European Regional Development Fund (ERDF). IMT Nord Europe participated in the COST COLOSSAL Action CA16109. The ATOLL site is one of the French ACTRIS National Facilities and contributes to the CARA program of the LCSQA funded by

the French Ministry of Environment. A. Velazquez Garcia's PhD grant was supported by CONACYT (grant 2019-000004-01EXTF-00001) and the Hauts-de-France Regional Council.

**Acknowledgments**

The authors are highly grateful to the staff at LOA (P. Goloub, F. Auriol, R. De Filippi, E. Bourrianne) and CERI EE (E.

Tison) for supporting the technical and logistical implementation of the instruments. We thank T. Podvin (LOA) and A. Bourin (CERI EE) for providing the weather data at ATOLL, and retrieving the 1-hour backtrajectories, respectively.

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
