# Peer review of "Assessment of light-absorbing carbonaceous aerosol origins and properties at the ATOLL site in Northern France"

_Aerosol Research, 2024_

## Author Comment (AC1)

The authors appreciate the effort from the reviewer and acknowledge his/her contribution to improving this manuscript. The replies are provided in red.

**Rewiever 1**

The manuscript show the results obtained with a new methodology: to apportion source contributions (shipping, vehicular, residential heating, industrial) and study optical properties variation of BrC species due to atmospheric aging. This methodology, named INTERPLAY, integrates multi-year aerosol observations with air mass back-trajectories and emission inventories.

The paper is very good, showing interesting and sounding results that can be useful at different scientific and technical levels. The manuscript very well written and clear in every part.

Thank you.

**General comment**: the results obtained following the INTERPLAY approach are well discussed, with a focus on BrC lifetime, as well as the comparison with the output of the Aethalometer model. However, the manuscript lacks sufficient information on the model itself, effectively making it impossible for another researcher to attempt to replicate it in another context. While this is understandable (given that this is a research article), my personal position is that this powerful tool should (anyway) be better described in 2.5, with the basic technical details in the supplement.

Following the remark from the reviewer, we decided to place the INTERPLAY code on the SI. This will allow reproducibility while keeping the methodology section with enough details for the aerosol community.

**Specific comments**:

Line 14: I would explain also in the abstract what "ATOLL" stands for.

Done.

Line 31: Why is "Light Absorbing Carbonaceous particles" acronym written in capital letters (LAC) while "Aerosol-Radiation Interaction" acronym (ari) in lowercase? Same for aci.

We agree with the reviewer, to be consistent, "aerosol-radiation interaction" and "aerosol-cloud interaction" are now all capitalized.

Line 308: do the Authors intend "AAEff" and "AAEwb" instead of "BCff" and "BCwb"? I suspect there is some confusion here. BC, optically speaking, is something with the AAE close to 1 by definition, and the deviation from this value depends mainly on the mixing/coating state/size distribution. Other things are BC from fossil fuel combustion (BCff) and by wood burning (BCwb). Together they form BC (BCff + BCwb = BC). But BCwb has an AAEwb close to 1 since it is still BC. In the literature are reported values of AAEwb up to 11, but (as done in the present paper) a value of 2 is generally accepted. But this value (AAEwb) is not characteristic of BCwb alone nor of the BrC present in the PM produced by wood combustion: it is an "effective average value" between $AAE_{BCwb}$ and $AAE_{BrC}$, as well as the most critical parameter to be set in the Aethalometer model. Please take care of the differences between absorption coefficients apportioned by the Aethalometer model (babsff, babswb), related masses BCff, BCwb, and spectral dependencies of the light absorbing species AAEff,

AAE$_{wb}$, AAE$_{BC}$ and AAE$_{BrC}$ and consider revising this part. I actually suggest having a look at the following paper: Multi-wavelength optical determination of black and brown carbon in atmospheric aerosols, https://doi.org/10.1016/j.atmosenv.2015.02.058.

As correctly identified by the reviewer, there is a typo in the manuscript, and we were of course referring to the associated absorption angstrom exponents (AAE$_{ff}$ and AAE$_{wb}$) instead of different components of BC according to the aethalometer model (BC$_{ff}$ and BC$_{wb}$). This has been corrected.

---

## Author Comment (AC2)

The authors appreciate the effort from the reviewer and acknowledge his/her contribution to improving this manuscript. The replies are provided in red.

**Reviewer 2**

The manuscript titled "Assessment of light-adsorbing carbonaceous aerosol origins and properties at the ATOLL site in Northern France" apportions, by using the INTERPLAY integrated multi-step approach (with back trajectories and emission inventories, in-situ measurements), sources (shipping, vehicular, residential heating, industrial) of BC and BrC and their lifecycles, focusing on effect of aging processes on optical properties of BrC.

The manuscript is well structured and written, the research question is properly outlined and clearly addressed. Also, the methodology is exhaustively explained and consistent with the main objectives. References are appropriate and key studies included. The topic and the submitted study is very interesting, for the experimental integrated approach, and complete in discussing obtained results.

In my opinion, the study is valuable and could have a very good research sound for the research community, needing only some little refinements.

Thank you.

**General comments**

The applied approach is innovative, with uncertainties discussed in the conclusions, and easily scalable, being based on available/easily accessible data.

Apart from mentioning previous studies performed at ATOLL site and in Paris, are there other European sites with same measurements types to be compared? Please, report some of them and discuss results. The authors could have a look at these papers, about other sites belonging to the ACTRIS network: https://doi.org/10.1016/j.atmosres.2017.10.004 & https://doi.org/10.1016/j.atmosres.2020.104976

The following sentence has been added including the reference for a recent review on multiple European sites on L.201:

"Those values are within other European cities, with BC (i.e. the sum of $BC_{ff}$ and $BC_{wb}$) ranging from 0.7 to 1.7 µg m-3, generally following an increasing trend from north to south (Savadkoohi et al., 2023)."

Line 163: according to which criterion is a minimum contribution of 20% of the total integrated BC chosen? Please provide references, in case some studies applied it before.

The threshold of 20% was used to find a balance between a representative number of points (at least 70 back trajectories) for the main sources. Increased percentage (e.g. 30%) has strongly affected the statistics without any meaningful impact on the obtained results for the most abundant sources. The test of the thresholds has been added in the supplementary information (Table S1).

**Minor revisions**

- The first and last statements of the abstract are almost equal. Delete one of them.

We propose to modify the last sentence, focusing particularly on the region of study, as:

"The results from this study allows for an improved understanding of sources and atmospheric dynamics of light-absorbing carbonaceous aerosols in northern France, being crucial for both source abatement strategies as well as a better assessment of their climate impact."

- Regarding lifecycle of LAC, take a look at "Liu, D., He, C., Schwarz, J.P. et al. Lifecycle of light-absorbing carbonaceous aerosols in the atmosphere. npj Clim Atmos Sci 3, 40 (2020). https://doi.org/10.1038/s41612-020-00145-8"

We thank the reviewer for suggesting the inclusion of this interesting and updated review about LACs. It is now part of the references in the main text.

- Check citation style throughout the text (i.e., missing commas).

The correction has been done.

- Line 124: why are not world-historical trend of emission inventory extended until 2019 (study period)?

The latest version of EDGAR only covers up to 2018, indeed not fulfilling our entire period, but without any meaningful expected changes in 2019.

- Does France include Lille or not in the pie-charts of the Figures 2b and 3a, 3b?

No, France includes the contribution of the whole territory except for the contribution of Lille. This specification has been added to the description in figures 2b, 3a, 3b.

"...note that France does not include Lille's contribution..."

- Line 260: Figure S7 is wrongly indicated as Figure S5.

The correction has been done.

- Line 335-336: sentence to be re-phrase.

The sentence between 335-336 was modified as following:

"This fraction is comparable to Athens (24%), despite their higher total absorption at 370 nm (15.9 Mm$^{-1}$) (Liakakou et al., 2020), and marked differences in weather conditions and heating fuel usage compared to Lille (Rehfeldt et al., 2020)."

- Line 371-372: re-phrase the sentence for missing verb.

The sentence has been corrected as such:

"Relative contributions from residential and traffic emitted less than 24 hrs (fresh) and more than 24hrs (aged) before reaching the ATOLL site **were analyzed concerning**: (a) the mass loading of BC; (b) the light absorption coefficient of BrC calculated at 470 nm..."

- Figure S1: check incorrect figure numbering and caption.

The numbering and caption have been corrected accordingly.

- Figure S7 could be moved as Figure S6.

The correction has been done.